# In-Depth Understanding of the Chemical Stripping Mechanism of AlSiY Coatings on Nickel Superalloys by First-Principles Calculation

Hongying Li [1,2,*], Chaoyong Luo [2], Ce Zhang [2], Lei Wu [2], Xiaolong Zhou [3], Chengsong Zhang [3], Yang Wang [2] and Zhiwu Wang [1,*]

1   School of Power and Energy, Northwestern Polytechnical University, Xi'an 710072, China
2   AECC Aero Science and Technology Co., Ltd., Chengdu 610503, China; luochaoyong123@163.com (C.L.); zctyson14@163.com (C.Z.); wulei@scfast.com (L.W.); 19141320872@163.com (Y.W.)
3   School of Materials Science and Engineering, Southwest Jiaotong University, Chengdu 610036, China; zxl001227@my.swjtu.edu.cn (X.Z.); cszhang@swjtu.edu.cn (C.Z.)
*   Correspondence: hylicd@sina.com (H.L.); malsoo@mail.nwpu.edu.cn (Z.W.)

**Abstract:** In order to repair a failed AlSiY coating on aeroengine turbine components, the AlSiY coating was stripped using a nitric acid-based removal reagent. The homogeneity of chemical stripping was evaluated and the stripping mechanism was clarified using first-principles calculations. The effects of hydrofluoric acid (HF) and chromium trioxide ($CrO_3$) on the homogeneity of chemical stripping were investigated by calculating the electronic work functions (EWFs) of the stripped surfaces. The results showed that the stripped surfaces of the AlSiY coating exhibited a serrated appearance when it was etched by a single nitric acid solution, indicating severely inhomogeneous stripping. With the addition of HF and/or $CrO_3$, the homogeneity of chemical stripping could be greatly improved, which was attributed to the increased EWF of the (200) surface and the decreased EWF of the (110) surface, causing the corrosion cathode to transition from the (110) surface to the (200) surface. The $HF+CrO_3+HNO_3$ mixed reagent was the optimal combination for the uniform stripping of the AlSiY coating, while the inner layer was not broken. The Al atoms on the surface could be preferentially removed due to the strong bonding with acid ions. The research method proposed in the present work will provide a new means to design chemical removal reagents.

**Keywords:** chemical stripping; mechanism; AlSiY coating; nickel-based superalloy; first-principles calculation





## 1. Introduction

Nickel-based superalloys are usually used for the manufacturing of aeroengine turbine components, which always display gas erosion at high temperatures [1]. To extend the service life of turbine components, aluminide coatings, such as Al-Si and Al-Si-Y coatings, are usually employed to prevent the degradation of the components [2–4]. During service, the aluminide coatings gradually degrade due to the depletion of Al in the coatings, as well as the consequent phase transformation from β-NiAl to γ′-Ni$_3$Al [5]. Although the addition of Pt could improve the hot corrosion resistance of aluminide coatings by promoting the formation of α-Al$_2$O$_3$ scales [6], as well as improving the spallation resistance of α-Al$_2$O$_3$ scales [7,8], the aggressive service environment inevitably leads to the rapid degradation of aluminide coatings. Littner's research pointed out that when the Al content in the coating is depleted to below 30 at. %, the protectiveness of the aluminide coating will be lost [7]. Because of the high cost of turbine components, the repair of failed aluminide coatings is more economical than replacing them with new components. Thus, the refurbishment of degraded aluminide coatings attracts much attention in aerospace manufacturing. Throughout the whole remanufacturing process, the stripping of aluminide coatings is a critical step

that provides a fine surface for coating refurbishment. The stripping quality determines the service life of refurbished aluminide coatings. At present, the stripping technologies of aluminide coatings mainly focus on mechanical and chemical methods. For mechanical stripping, grit blasting is the most commonly used method. However, grit blasting is only suitable for the removal of the residual oxides on the surfaces of aluminide coatings [9,10]. For Al diffusion coatings, mechanical stripping appears ineffective. Chemical stripping, including electrochemical stripping, is still the most effective way to remove Al diffusion coatings. For chemical stripping, the combination of acids is the key factor for uniform stripping. On the other hand, such a combination of acids should be safe for the surfaces of substrates. Considering the requirements above, many acids and inorganic reagents, such as $(NH_4)_6Mo_7O_{24}$, $HNO_3$, HF and HCl, and their combinations, have been adopted [11–14]. Poupard et al. investigated various soft chemical stripping methods and found that chemical baths containing relatively high concentrations of pitting reagents in organic media are the most effective [15]. It is worth noting that the application scope of chemical removal reagents is limited. For a new aluminide coating, a special chemical removal reagent usually needs to be redeveloped by adjusting the ratio and combination of acids. Although many novel chemical removal reagents have been developed successfully, few reports discuss the function of each acid in the chemical removal reagent. Considering the research gap above, the present work will investigate the uniform stripping of the AlSiY coating on the K465 superalloy using a nitric acid-based removal reagent. The electronic work functions of the stripping surfaces and the stripping mechanism of the AlSiY coating are calculated by the first-principles method, and the effects of the addition of hydrofluoric acid (HF) and chromium trioxide ($CrO_3$) on the uniform stripping of the AlSiY coating are clarified. The results will guide the design of new chemical removal reagents.

## 2. Experimental and Calculation Details

### 2.1. Preparation of Aluminide Coatings

The cast K465 nickel-based superalloy was selected as a substrate with the chemical concentrations of 9.90 W—9.5 Co—8.66 Cr—5.10 Al—2.52 Ti—1.44 Mo—0.95 Nb—0.16 C—0.019 B—0.037 Zr—balance Ni (in wt.%). The samples were machined to the size of $20 \times 10 \times 1$ mm$^3$, and then the AlSiY coating was deposited on the surface of the K465 superalloy by multi-arc ion plating for 60 min. After deposition, the coated samples were heated at 1050 °C for 180 min and a diffused AlSiY coating was formed on the surface of the K465 superalloy.

### 2.2. Chemical Stripping Procedures

To evaluate the effects of removal reagents on the uniform stripping of the AlSiY coating, four removal reagents containing different proportions of nitric acid ($HNO_3$), hydrofluoric acid (HF) and chromium trioxide ($CrO_3$) were compared. Their concentrations are listed in Table 1. The coated samples were immersed in the removal reagents for 30 min at room temperature. After chemical stripping, the stripped samples were ultrasonically cleaned with ethyl alcohol (AR) for microstructural observation.

**Table 1.** The concentrations of removal reagents used in the present work (g/L).

| Combinations | HNO$_3$ | CrO$_3$ | HF |
|---|---|---|---|
| HNO$_3$ | 120 | - | - |
| CrO$_3$+HNO$_3$ | 120 | 8 | - |
| HF+HNO$_3$ | 120 | - | 11 |
| HF+CrO$_3$+HNO$_3$ | 120 | 8 | 11 |

### 2.3. Characterization

To provide structural information for first-principles calculations, the coated samples were characterized by microstructural observation and phase composition measurement.

A cross-section of the coated sample was prepared by mechanically grinding and polishing it for microstructure observation. The cross-section was etched with 4 vol.% Nital reagent and observed via scanning electron microscopy (SEM, JEM-IT500, Japan Electron Optics Laboratory Co., Ltd., Mitaka, Japan). The elemental distribution in the AlSiY coating was measured by an energy-dispersive X-ray analyzer (EDS, equipped with SEM). The phase composition of the AlSiY coating was identified by X-ray diffraction measurement (XRD, X'Pert Pro, Malvern Panalytical, Almelo, The Netherlands) with Cu-K$\alpha$ radiation. The diffraction angle was in the range of 20–100° with a scanning step of 0.02°. Additionally, the cross-sections of stripped samples were observed using optical microscopy (OM, Olympus GX51F, Olympus Corporation, Tokyo, Japan) to evaluate the flatness of the stripped surfaces.

### 2.4. First-Principles Calculation

In order to clarify the effects of the removal reagent components on the uniform stripping of the AlSiY coating, the electronic work functions (EWFs) of the stripped surfaces were calculated in the framework of first-principles calculation. According to the phase composition measured by XRD, the β-NiAl phase dominated in the AlSiY coating and there were four main crystal orientations on the stripped surfaces, i.e., (110), (100), (200) and (211). Thus, the chemical reactions between the removal reagents and stripped surfaces were systematically investigated on these four surfaces. In the present work, three removal reagent components were considered, i.e., nitric acid ($HNO_3$), hydrofluoric acid (HF) and chromium trioxide ($CrO_3$). Corresponding to the experiment, four different combinations of acid ions were imported onto the stripped surfaces. They were $HNO_3$-only, a $CrO_3+HNO_3$ mixture, a $HF+HNO_3$ mixture and a $HF+CrO_3+HNO_3$ mixture. Additionally, different amounts of $HNO_3$ in the acid mixtures were considered. For the $HNO_3$-only reagent, a $3 \times 3$ surface slab model containing 1 to 3 nitrate ions ($NO_3^-$) was built. For the mixture reagents, one fluorinion ($F^-$) or/and a chromium trioxide molecule was added to the surface slab model. Then, the surface slab models were optimized and the EWFs and electronic structures of the stripped surfaces could be obtained. Considering the complex structure of liquid removal reagents, the adsorption sites of acid ions on the surfaces are usually randomly distributed. Thus, in the present work, the acid ions were also randomly added on the specific adsorption sites of the calculated surface. At least four different configurations for each condition were calculated and the average EWF was applied to evaluate the stripping tendency of the stripped surfaces. The calculated surface slab models and the adsorption sites are schematically shown in Figure 1.

All calculations were performed on the CASTEP code based on density function theory (DFT) [16]. The generalized gradient approximation (GGA) in the Perdew–Burke–Eruzerh (PBE) formula was used for the exchange-correlation potential [17]. The electronic interaction was evaluated by ultrasoft pseudopotentials [18]. The crystalline structure of β-NiAl was built according to the description in the reference [19]. The calculated surfaces were cleaved from the bulk structure of the β-NiAl phase. The cut-off energy was set to 450 eV. The k-point grids were set according to the lattice parameters: $2 \times 2 \times 1$ for the β-NiAl (110), β-NiAl (100) and β-NiAl (200) slabs; $1 \times 2 \times 1$ for the β-NiAl (211) slab. During the geometry optimizations, the convergence criteria of energy, force, stress and displacement were set as $5 \times 10^{-5}$ eV/atom, 0.1 eV/Å, 0.2 GPa and 0.005 Å, respectively.

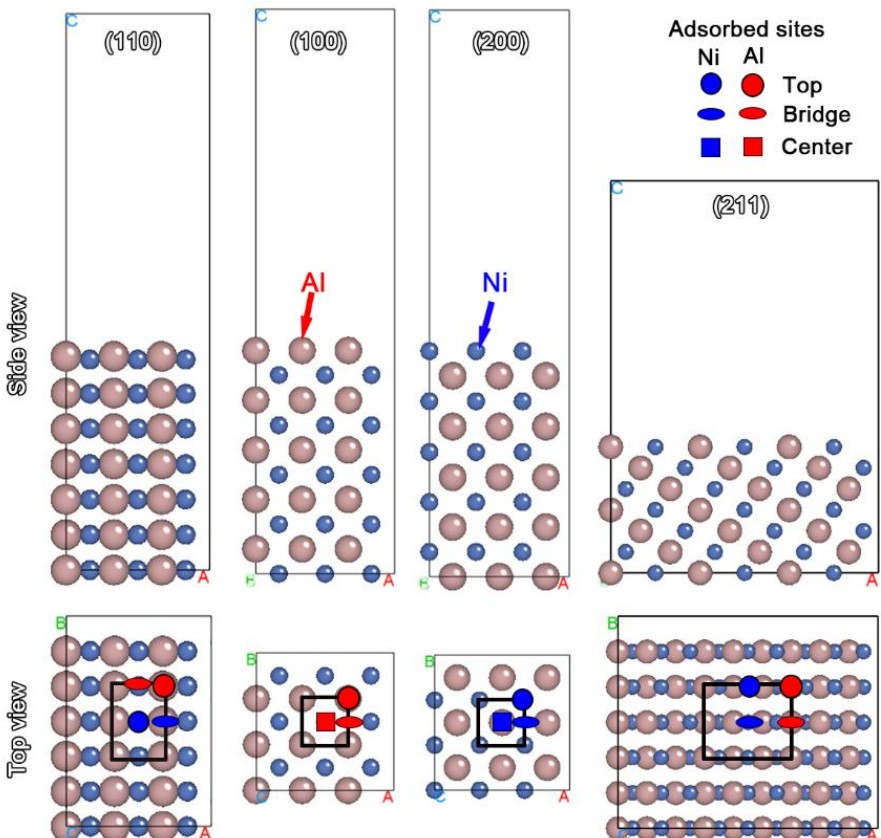

**Figure 1.** The calculated surface slab models and the adsorption sites for $F^-$, $NO_3^-$ and $CrO_3$. The (110) and (211) slabs contain 7 atomic layers and the (100) and (200) slab contain 10 atomic layers. The thicknesses of the vacuum layers for (110), (100), (200) and (211) slabs are 20, 20, 20 and 15 Å, respectively. The adsorption sites are marked in the figures. The circle, ellipse and square denote top, bridge and center sites, respectively. The color distinguishes different atoms. Blue and red indicate Ni and Al atoms, respectively.

## 3. Results and Discussion

### 3.1. Microstructure of the AlSiY Coating

Figure 2 shows the SEM image of the cross-sectional morphology of the AlSiY coating plated on the K465 superalloy. It can be clearly seen that the AlSiY coating is composed of two layers. The thickness of the outer layer and the inner layer is about 30 and 15 μm, respectively. To clarify the difference between these two layers, the elemental distribution was measured by EDS. The whole AlSiY coating is formed by interdiffusion between the deposited AlSiY coating and the substrate. The elements, Ni, Co, W, Cr and Ti diffuse outward, while Al, Si and Y diffuse inward. There exists a concentration difference between the two layers after diffusion. The alloy elements, such as Co, W, Cr and Ti, are mainly concentrated in the inner layer, while Al is mainly distributed in the outer layer. The high concentrations of Ti, W and Cr in the inner layer correspond to the high content of carbides.

Further identifying the phase composition of the AlSiY coating, the depth-related XRD patterns of the AlSiY coating were tested, as shown in Figure 3. It can be seen that both layers mainly consist of the β-NiAl phase and traces of $M_{23}C_6$ and $M_6C$ carbides. Additionally, the content of carbides in the inner layer is higher than that in the outer layer, which corresponds to the enriched alloy elements in the inner layer. It is worth noting that the diffraction peaks of the β-NiAl phase for the outer layer shift to a higher diffraction angle as compared with that for the inner layer. This indicates the high solubility of the alloy elements in the β-NiAl phase, resulting in lattice expansion.

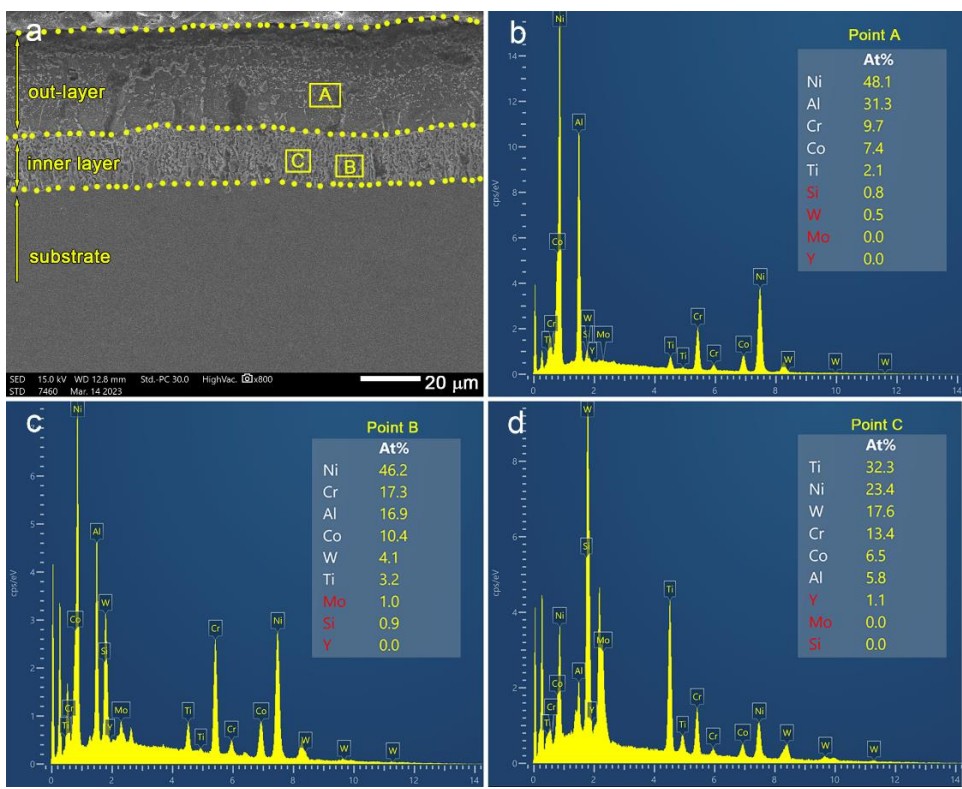

**Figure 2.** SEM image of cross-sectional morphology of the AlSiY coating deposited on K465 superalloy and the elemental distribution in the deposited AlSiY coating. (**a**) SEM image of cross-sectional AlSiY coating; EDS spectra of (**b**) point A, (**c**) point B and (**d**) point C in (**a**).

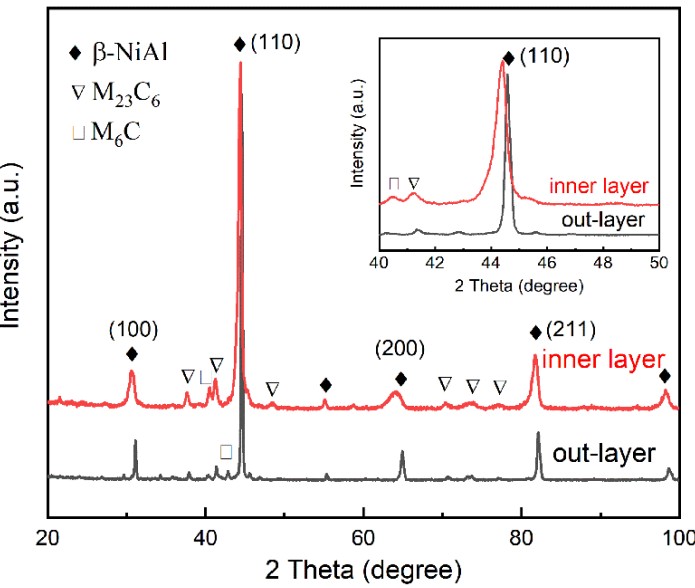

**Figure 3.** Depth-related XRD patterns of the deposited AlSiY coating. The enlarged XRD patterns ranging from 40° to 50° are shown in the inset.

### 3.2. Microstructure of the AlSiY Coating after Chemical Stripping

Figure 4 shows the cross-sectional morphologies of the AlSiY coating after chemical stripping. It can be seen that there exist obvious etch pits in the AlSiY coating for all removal reagents. The removal rate depends on the combination of removal reagents, which is in the order of $CrO_3+HNO_3 < HF+HNO_3 < HF+CrO_3+HNO_3$. Additionally, the size of the etch pits also depends on the combination of removal reagents. For the $CrO_3+HNO_3$

mixture-etched coating, the pit is small and shallow, while, for the HF+HNO$_3$ mixture-etched coating, the pit is large and deep. This indicates that the etched homogeneity of the HF+HNO$_3$ mixture is lower than that of the CrO$_3$+HNO$_3$ mixture. It seems that the HF+CrO$_3$+HNO$_3$ mixture displays the worst homogeneity among all removal reagents due to its serrated surfaces after stripping (see Figure 4d). It should be noted that the serrated appearance can be only found in the inner layer. However, the smooth surface still appears in the outer layer (see Figure 4e). There is a great difference in the alloy element content between the outer layer and the inner layer, which indeed causes changes in the etched appearance. To evaluate the homogeneity of the stripped surfaces, the cross-sectional area of the etch pits in one metallographic image was calculated. The higher the homogeneity, the smaller the area. Several images were obtained and the average area was determined. Figure 5 shows the statistics of the area of the etch pits for different removal reagents. It shows that the CrO$_3$+HNO$_3$ mixture seems to be the best choice for the chemical stripping of aluminide coatings due to the lowest average area of etch pits and smoothest stripped surfaces, as shown in Figure 4b. However, it should be noted that the average area of the etch pits represents the homogeneity of chemical stripping in the microscopic scale. The statistical distribution of the etch pit area indicates the macroscopic homogeneity. The large gap in the distribution of the etch pit area for the CrO3+HNO$_3$ mixture indicates its inhomogeneity in the macroscopic scale. By this consideration, the CrO$_3$+HF+HNO$_3$ mixture should be the best choice if the stripping time is accurately controlled to avoid the stripping of inner layers.

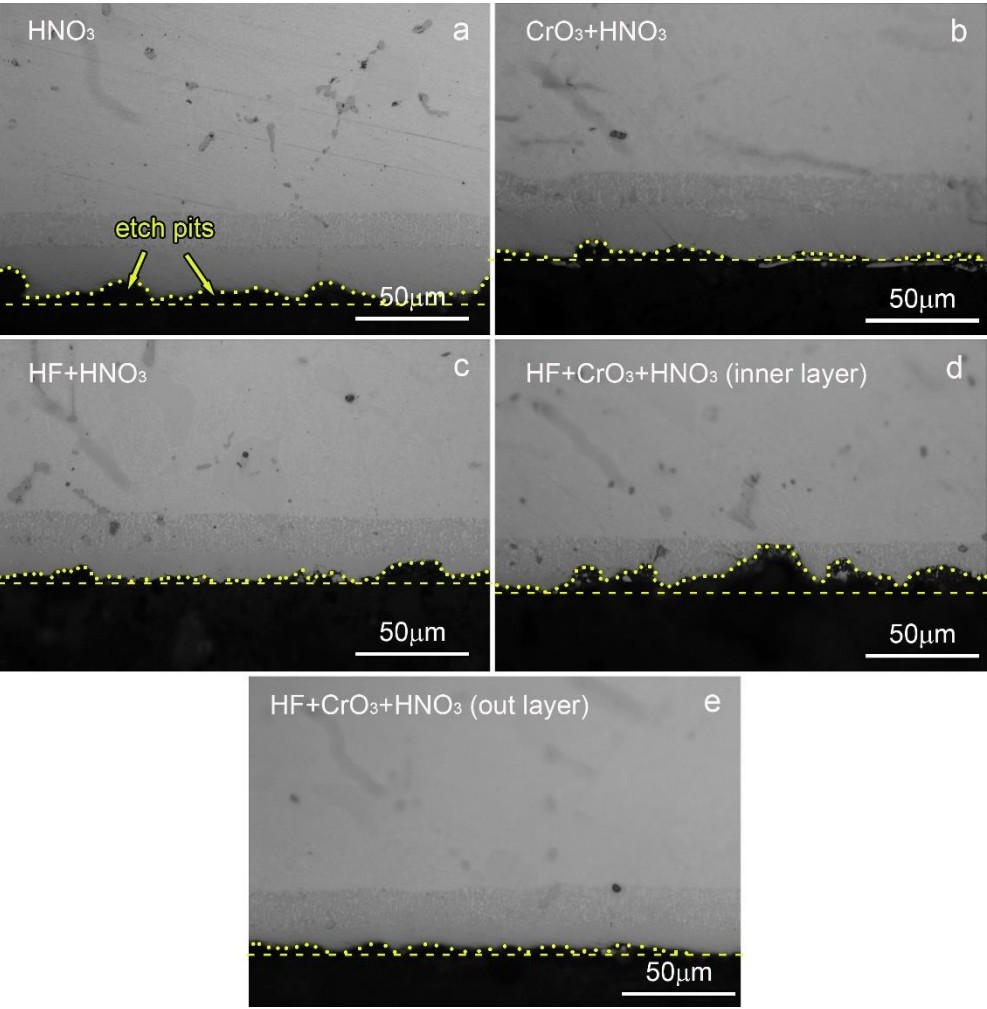

**Figure 4.** Cross-sectional morphologies of the AlSiY coatings after chemical stripping using (**a**) HNO$_3$-only, (**b**) CrO$_3$+HNO$_3$ mixed and (**c**) HF+HNO$_3$ mixed removal reagents. (**d**,**e**) show

the inner layer and smooth outer layer after chemical stripping using the HF+CrO$_3$+HNO$_3$ mixed removal reagent, respectively. The dashed lines represent the profiles of smooth surfaces and the dotted lines represent the profiles of etched surfaces. The area enclosed by the dashed lines and dotted lines and its shapes show the homogeneity of chemical stripping.

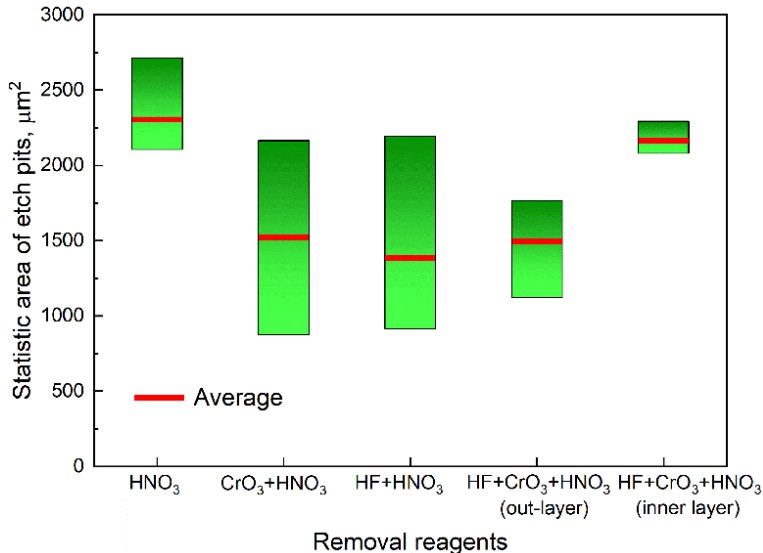

**Figure 5.** The statistical area of etch pits for different removal reagents. A low average value indicates the good homogeneity of chemical stripping in the microscopic scale, and a narrow statistical distribution indicates good homogeneity in the macroscopic scale.

### 3.3. Explanation of Uniform Stripping by First-Principles Calculation

To clarify the relationship between the morphology of stripped surfaces and the combination of removal reagents, the electronic work functions (EWFs) of the stripped surfaces are calculated by the first-principles method. The EWF represents the energy difference between the vacuum level and the Fermi level. The chemical stripping process can be considered as a redox reaction that occurs on the surface of the AlSiY coating. The higher the EWF is, the harder it is to oxidize [20]. Additionally, there exists galvanic corrosion between surfaces with different EWFs. The electrons will transfer from the surfaces with high EWF to the ones with low EWF. The surfaces with high EWF will act as cathodes during the redox reaction, while the surfaces with low EWF will be the anodes and be corroded. Figure 6 gives the calculated EWFs of the stripped surfaces etched by different removal reagents containing different nitric acid content. For the HNO$_3$-only reagents, the (110) surface has the highest EWF among all stripped surfaces. With the increase in HNO$_3$ content, the EWF of the (200) surface increases and is close to that of the (110) surface. It should be noted that the (110) surface dominates the whole surface of the AlSiY coating (see Figure 3). In this condition, the area of the cathode ((110) surface) is larger than that of the anode (other surfaces besides (110)). Severe pit corrosion will occur on such small anode surfaces, which corresponds to the etched pits as shown in Figure 4a. For a single acid solution, such as HF or CrO$_3$, severe pit corrosion cannot be avoided effectively (see Figure 6b,c). Therefore, mixed acid solutions are considered. When a small amount of CrO$_3$ or HF is added to the HNO$_3$ solution, there is no obvious change in the EWF, except for the EWF of the (200) surface (see Figure 6b,c). As compared with HF addition, CrO$_3$ addition greatly increases the EWF of the (200) surface, which even surpasses the EWF of the (110) surface at high HNO$_3$ content conditions. This indicates that the cathode has changed from the (110) surface to the (200) surface. The area of the anode is greatly increased and shallow pits are observed for the stripped surface treated with the CrO$_3$+HNO$_3$ mixture (see Figure 4b). It is worth noting that the HF+CrO$_3$ mixed addition shows the greatest

enhancement in the EWF of the (200) surfaces. When the $HNO_3$ content is still at a lower level, the EWF of the (200) surfaces reaches a high value. Most importantly, the difference in the EWFs among the (110), (100) and (211) surfaces decreases, which is beneficial for the uniform stripping of the AlSiY coating. Additionally, it is found that the average EWF of the (110) surfaces treated with the $HF+CrO_3+HNO_3$ mixture is lower than for those treated with other removal reagents, indicating the high removal rate obtained when using the $HF+CrO_3+HNO_3$ mixture. This is also proven by the microstructural observation (see Figure 4d). It should be pointed out that all EWF calculations were conducted on a pure β-NiAl phase, without considering the effects of alloy elements. If the chemical stripping occurs in the inner layers, which contain massive alloy elements, the stripping homogeneity will be changed and the effects of the alloy elements on the EWFs should be considered. According to the calculated results, the $HF+CrO_3+HNO_3$ mixture is considered the best choice for the uniform stripping of the AlSiY coating.

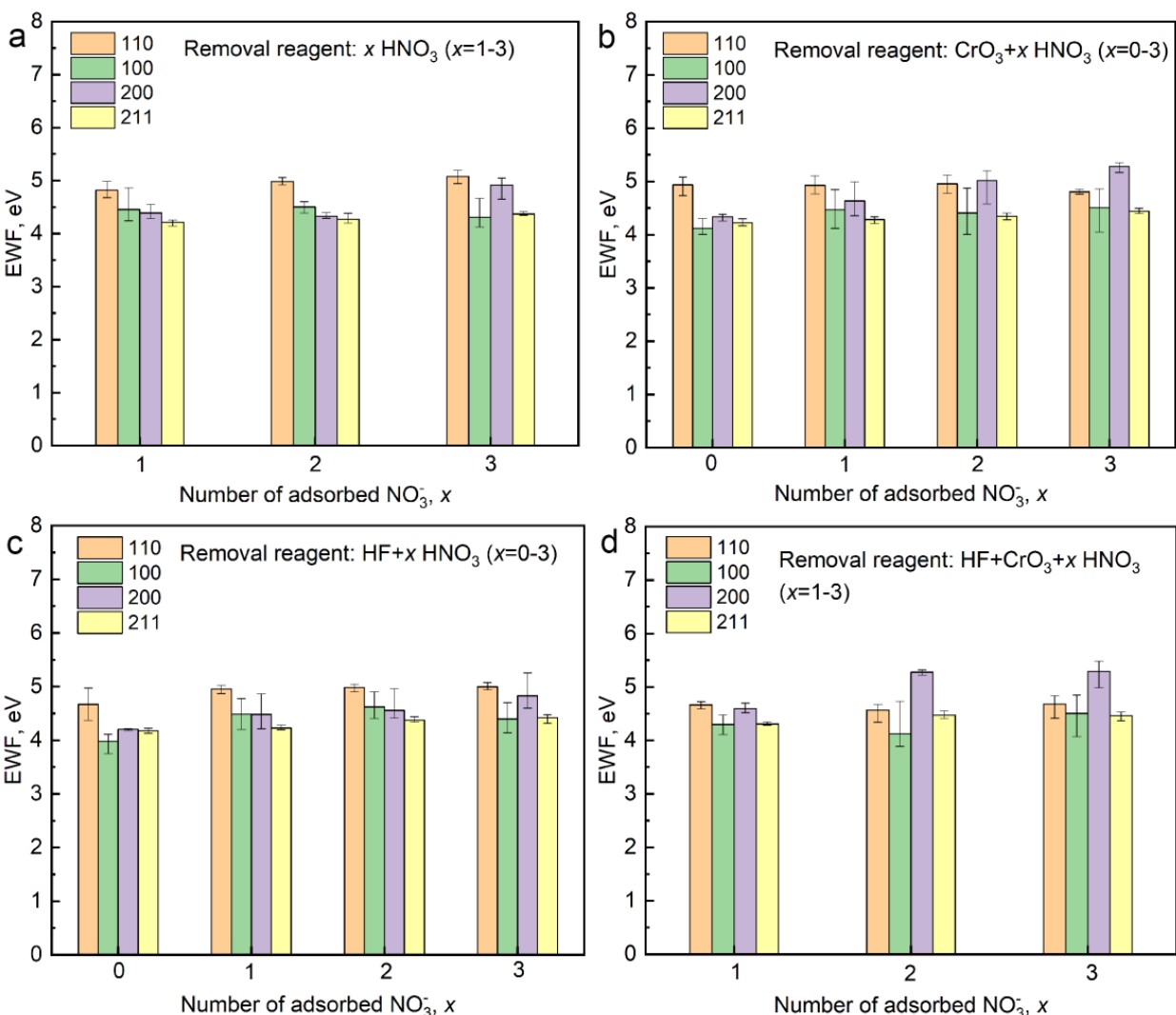

**Figure 6.** Electronic work functions (EWFs) of stripped surfaces treated with (**a**) $HNO_3$-only, (**b**) $CrO_3+HNO_3$ mixture, (**c**) $HF+HNO_3$ mixture and (**d**) $HF+CrO_3+HNO_3$ mixture containing different nitric acid content. The stripped surfaces with high EWFs will act as cathodes during chemical stripping.

### *3.4. Stripping Mechanism of Aluminide Coatings*

To explore the stripping mechanism of the AlSiY coating, the bonding characteristics of atoms on the surfaces should be identified. Figure 7 shows the optimized configurations

of the (110) surface with an adsorbed acid ion. It can be seen that the Al atoms close to the oxygen or fluorine ions seem to be pulled out from the surface. This indicates that the oxygen or fluorine ions in the acid solutions prefer to strip the Al atoms. The Ni atoms will subsequently be removed due to the breakage of the Ni-Al bonds. To further clarify the bonding characteristics on the stripped surfaces, the bond population was analyzed, as listed in Table 2. Generally, the higher the bond population, the stronger the bond [21]. It can be found that all bonds are mixed types containing an ionic bond and covalent bond. The ionic component dominates all bonds. The bond strength depends on the absorbed sites of ion species. Generally, the population of Al-O bonds is higher than that of Ni-O bonds, while the population of Al-F bonds is close to that of Ni-F bonds. When the acid ions are adsorbed on the surface, the O or F ions prefer to bond with Al due to the tight bonds. Thus, the removal rate of the AlSiY coating is mainly determined by the bonding strength between metal atoms and acid ions.

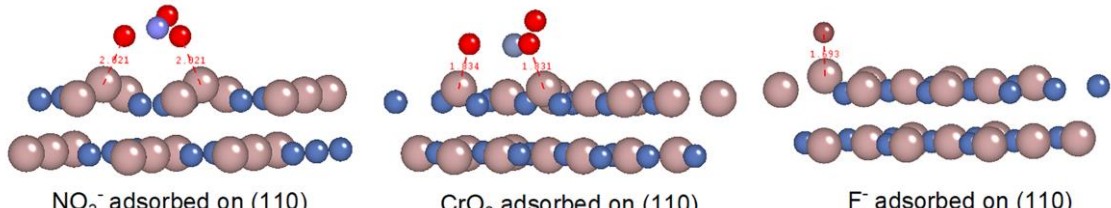

**Figure 7.** The optimized configurations of the (110) surface with an adsorbed acid ion. The Al atoms in the top surface seem to be pulled out from the surface, indicating that the oxygen or fluorine ions prefer to strip the Al atoms.

**Table 2.** Calculated bond populations of stripped surfaces adsorbed by acid ions.

| Surface | Adsorbed Site | Ion Species | | | | | |
| --- | --- | --- | --- | --- | --- | --- | --- |
| | | $NO_3^-$ | | $F^-$ | | $CrO_3$ | |
| | | Bonding Type | | | | | |
| | | Al-O | Ni-O | Al-F | Ni-F | Al-O | Ni-O |
| (110) | Ni-top | 0.3 | - | - | 0.47 | 0.47 | - |
| | Al-top | 0.46 | - | 0.49 | - | 0.43 | 0.22 |
| | Ni-bridge | 0.46 | 0.34 | 0.04 | 0.16 | 0.45 | 0.34 |
| | Al-bridge | 0.39 | 0.39 | 0.22 | - | 0.48 | - |
| (100) | Al-top | 0.59 | - | 0.53 | - | 0.54 | - |
| | Al-bridge | 0.49 | - | 0.27 | - | 0.46 | - |
| | Al-center | 0.56 | - | 0.07 | - | 0.47 | - |
| (200) | Ni-top | - | 0.35 | - | 0.52 | - | 0.41 |
| | Ni-bridge | - | 0.39 | - | 0.31 | - | 0.45 |
| | Ni-center | - | 0.39 | - | 0.14 | - | 0.42 |
| (211) | Ni-top | 0.42 | - | - | 0.51 | 0.59 | - |
| | Al-top | 0.35 | 0.26 | 0.49 | - | 0.59 | - |
| | Ni-bridge | - | 0.29 | 0.30 | - | 0.48 | 0.27 |
| | Al-bridge | 0.43 | 0.38 | 0.24 | - | 0.45 | 0.3 |

Figure 8 shows the partial density of states (PDOS) of Al, Ni and O atoms on the stripped (110) and (200) surfaces with three nitrate ions' adsorption. It can be seen that all peaks of PDOS for O atoms overlap with those for Al and Ni atoms, indicating full hybridization among atomic orbits. The O-s electrons dominate in the energy range of −27.5 eV to 15 eV and bond with the s and p orbital electrons of metal elements. The O-p electrons bond with the s and p orbital electrons of metal elements in the energy range of −5 eV to the Fermi level. In addition, there are p-d bonding states between Al and Ni

atoms occupied in the energy range of −15 eV to the Fermi level. The great difference in the bonding characteristics of different stripped surfaces results in the inhomogeneous stripping of the AlSiY coating. For the (110) surface, the O-s orbit at −19.2 eV hybridized with Al atoms is more obvious than that with Ni atoms. This indicates that the Al-O bonds are stronger than the Ni-O bonds, which corresponds to the calculated bond populations. However, for the (200) surface, the O-s orbit (−26.8 eV and −21.6 eV) hybridized with Al and Ni atoms is weak. The bond hybridization between the O-p orbit and Ni atoms dominates the bonds on the (200) surfaces. The weakened bonds contribute to the improved anti-corrosion property on the stripped surfaces, as well as the increased EWF.

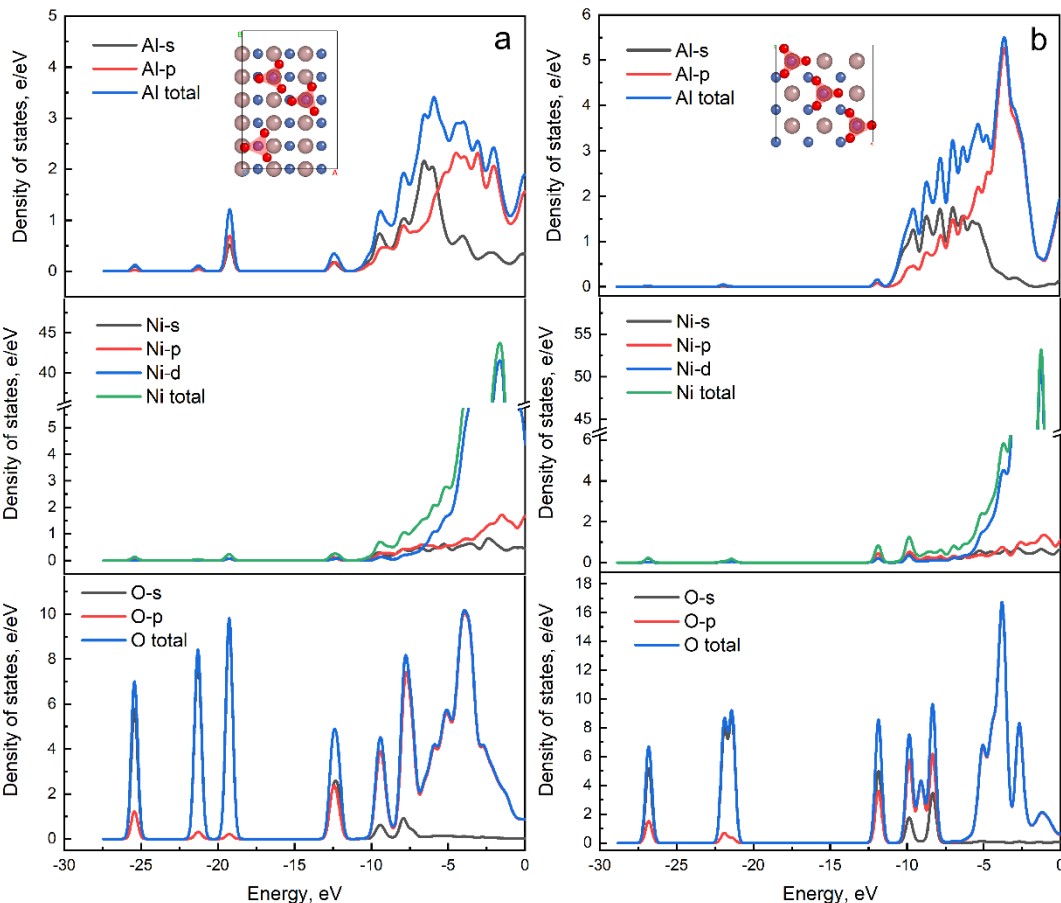

**Figure 8.** Partial density of states of Al, Ni and O atoms on the stripped (**a**) β-NiAl (110) and (**b**) β-NiAl (200) surface with three nitrate ions' adsorption. The displayed PDOSs correspond to the most stable adsorption configuration as shown in the insets. The Al and Ni atoms are selected from the topmost layers on the stripped surfaces.

Referring to the previous results, we found that the homogeneity of chemical stripping was improved when the HF and/or $CrO_3$ was added to the $HNO_3$ removal reagent. The calculated results of the EWFs explain this phenomenon via the change in the corrosion cathode from the (110) surface to the (200) surface. In fact, the EWFs on the stripped surfaces are significantly dependent on the bond characteristics. The number of electrons involved in bonding on the stripped surfaces is proportional to the corrosion tendency of the surfaces. Figure 9 shows the integrated density of states of the adsorbed O atoms on the (110) and (200) surfaces. Here, we only consider the O atoms in the $NO_3^-$ to clarify the effects of the HF and/or $CrO_3$ addition. For the (110) surface, the integrated DOS of adsorbed O atoms around the Fermi level increases with HF and/or $CrO_3$ addition, which indicates the enhanced corrosion rate on the (110) surface and the decreased EWF. Meanwhile, for the (200) surface, the opposite variation appears. The mixed removal reagent with HF

and $CrO_3$ addition has the highest integrated DOS on the (110) surfaces, as well as the lowest integrated DOS on the (200) surfaces, which ensures the cathode transformation between the (110) and (200) surfaces. The reduced area of the cathode is beneficial for the homogeneity of chemical stripping.

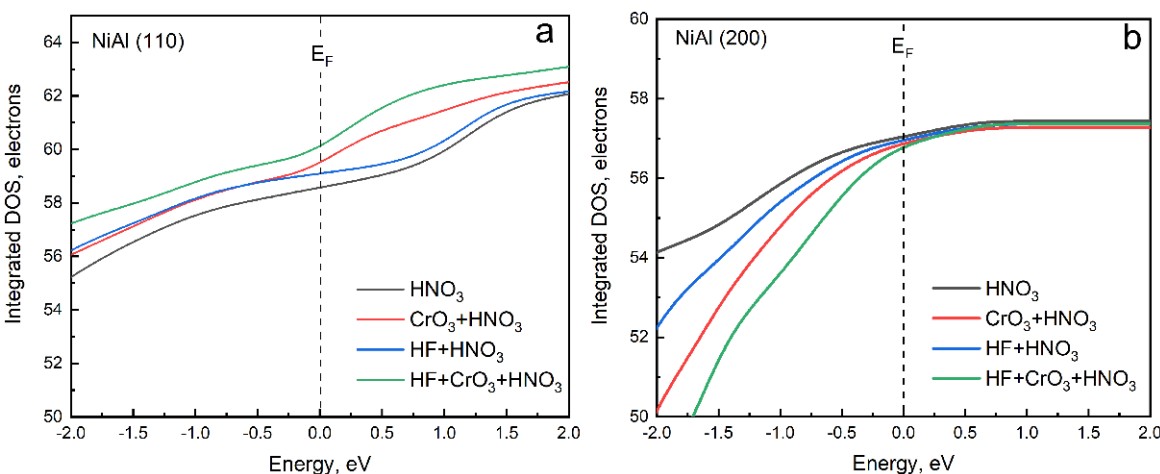

**Figure 9.** Integrated density of states of adsorbed O atoms on the (**a**) β-NiAl (110) and (**b**) β-NiAl (200) surfaces etched by different removal reagents. Only the O atoms in the $NO_3^-$ are considered.

## 4. Conclusions

The removal of a multi-arc ion deposited AlSiY coating was attempted using a nitric acid-based removal reagent. The first-principles calculations on the EWFs of the stripped surfaces were conducted to aid our understanding of the chemical stripping mechanism of aluminide coatings. The results are summarized as follows.

The multi-arc ion deposited AlSiY coating was composed of two layers. The alloy elements were mainly concentrated in the inner layer. Both layers consisted of β-NiAl phases and traces of $M_6C$ and $M_{23}C_6$ carbides.

Serrated corrosion pits appeared on the stripped surfaces of the AlSiY coating, indicating inhomogeneous stripping. The inhomogeneous stripping was mainly caused by the differences in corrosion rate for different stripped surfaces, which were determined by the EWFs of the surfaces.

HF and/or $CrO_3$ addition could improve the homogeneity of chemical stripping, which is attributed to the increased EWF of the (200) surface and the decreased EWF of the (110) surface, causing the corrosion cathode to transfer from the (110) surface to the (200) surface.

The $HF+CrO_3+HNO_3$ mixture reagent exhibited the best removal performance on the AlSiY coatings when the inner layer was not broken.

The acid ions preferred to strip Al atoms. Then, the Ni atoms would be removed due to the breakage of the Ni-Al bonds.

**Author Contributions:** Validation, C.Z. (Ce Zhang) and Y.W.; Formal analysis, C.L.; Investigation, C.Z. (Chengsong Zhang); Resources, Y.W.; Data curation, L.W. and Z.W.; Writing—original draft, H.L.; Writing—review & editing, C.Z. (Chengsong Zhang) and Z.W.; Supervision, X.Z. All authors have read and agreed to the published version of the manuscript.

**Funding:** This research was funded by the Natural Science Foundation of Sichuan Province of China through Grant No. 2022NSFSC1962, the National Natural Science Foundation of China through Grant No. 12372338, the Natural Science Foundation of Shaanxi Province of China through Grant No. 2022JZ-20.

**Institutional Review Board Statement:** Not applicable.

**Informed Consent Statement:** Not applicable.

**Data Availability Statement:** Data are contained within the article.

**Conflicts of Interest:** Authors Hongying Li, Chaoyong Luo, Ce Zhang, Lei Wu and Yang Wang were employed by the company AECC Aero Science and Technology Co., Ltd. The remaining authors declare that the research was conducted in the absence of any commercial or financial relationships that could be construed as a potential conflict of interest.

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
