# Peer review of "In-Depth Understanding of the Chemical Stripping Mechanism of AlSiY Coatings on Nickel Superalloys by First-Principles Calculation"

_coatings, doi:10.3390/coatings14010135_

Round 1

Reviewer 1 Report

Comments and Suggestions for Authors

Minor revision is suggested for the manuscript.
(1) The authors study is about nanofluid but very few related papers are cited, so the authors are
advised to read and cite the following papers in the introduction part.
https://dx.doi:10.3390/sym12101700, https://dx.doi.org/10.1038/s41598-020-80554-0, https://doi.org/10.1007/s13204-023-02981-5, https://doi.org/10.1007/s13204-023-02961-9, https://doi.org/10.1007/s13204-023-02976-2.
(2) Discuss briefly the type 1 and type 2 hot corrosion resistance of aluminide coating.
(3) Elaborate the mathematical calculations in sections 2.1, 2.2, 2.3, and 2.4.
(4) Discuss the performance and thermal stability of the present work.
(5) Provide more details about first principles simulation, concept and explanation.
(6) Provide the comparison with published work.
(7) Provide the Nomenclature.
Thanks

Comments on the Quality of English Language

 Minor editing of English language is required.

Author Response

Response to reviewer #1:     

First of all, thank you very much for reviewing our manuscript. Your comments are very significant and professional. In addition, it is very helpful to improve the quality of the manuscript to publish. According to your comments, we answer the following: 

Question 1: The authors study is about nanofluid but very few related papers are cited, so the authors are advised to read and cite the following papers in the introduction part.

https://dx.doi:10.3390/sym12101700, https://dx.doi.org/10.1038/s41598-020-80554-0, https://doi.org/10.1007/s13204-023-02981-5, https://doi.org/10.1007/s13204-023-02961-9, https://doi.org/10.1007/s13204-023-02976-2.

Answer 1:

Thank you for providing us so many references. After we carefully reviewed these references above, we found that it is not related to our investigation. Our present work aims to understand the chemical stripping mechanism of aluminide coating by a theoretical calculation rather to investigate the nanofluid. The inhomogeneous stripping is mainly attributed to the differences in corrosion rate on surfaces with various index. Here we use electronic work function to analyze the corrosion rate for different surfaces. By calculating the electronic work functions of stripped surfaces, the chemical removal reagents can be effectively designed. Although the references above are not related to our present work, they enriched our knowledge. Thanks a lot.     

Question 2: Discuss briefly the type 1 and type 2 hot corrosion resistance of aluminide coating.

Answer 2:

The type I hot corrosion is also called high-temperature hot corrosion which is characterized by the formation of a thick, porous, outer oxide layer. While the type II hot corrosion is called low-temperature hot corrosion which is characterized by the large oxide- and sulfide filled pits form. The most difference between the type I and type II hot corrosion is the service temperature and the microstructure of the outer oxide layer. The outer oxide layer can be easily removed by sand blasting. But the β-NiAl layer underneath the oxide layer should be removed by chemical immersion. In our present work, we investigate the chemical stripping of β-NiAl layer underneath the outer oxide layer. Therefore, it is not related to the type of hot corrosion.

Question 3: Elaborate the mathematical calculations in sections 2.1, 2.2, 2.3, and 2.4.

Answer 3:

In our present work, we mainly investigate the chemical stripping mechanism of aluminide coatings by first-principles calculations as well as experimental verifications. Therefore, section 2.1, 2.2, 2.3 and 2.4 mainly describe the conditions of samples, experiments, characterizations and first-principles calculations, respectively. The first-principles calculations were conducted on a software platform called Materials Studio. No mathematical calculations were involved in our present work.

Question 4: Discuss the performance and thermal stability of the present work.

Answer 4:

Our present work mainly focusses on the repairment of the failed aluminide coatings rather than investigating the performance of a fine aluminide coatings. For chemical stripping of aluminide coatings, the homogeneity is the key factor for the fine repairments. Therefore, we designed chemical removal reagents by first-principles calculations to achieve the homogeneous stripping.

Question 5: Provide more details about first principles simulation, concept and explanation

Answer 5:

The theory of first-principles calculations has been carefully described by Ref. [16]. All calculated details and key parameters were all listed in section 2.4. Other researchers could easily conduct the same calculations according to our given parameters. We added more descriptions of calculated slab models in the caption of Fig. 1.

Question 6: Provide the comparison with published work. 

Answer 6:

 It is the first time we use the first-principles calculation to explain the chemical stripping mechanism of aluminide coatings in the framework of electronic work function theory. There is no related data about both the calculation and the experiment can be found. Therefore, we conducted experiments to verify the correctness of our calculations. We think our present work is an innovative work.

Question 7: Provide the Nomenclature.

Answer 7:

There were no physical symbols in our present work except for the elemental symbols. Those elemental symbols are general symbols which will not cause confusion on understanding. Therefore, we think no needs to add a nomenclature. 

Reviewer 2 Report

Comments and Suggestions for Authors

referee report 

coatings-2812974-peer-review-v1

In-depth understanding of chemical stripping mechanism of aluminide coatings on nickel superalloys by first-principles calculation

Hongying Li et al.

This manuscript reports on an analysis of chemical stripping of the aluminide coatings on nickel superalloys. As this process is

important for manufacturing aeroengine turbine components, this is an important issue. The topic is well suited for Coatings.

The present manuscript comprises 9 figures, 2 tables and 21 references are given.

The present manuscript is well arranged, and all details of the processes applied are discussed properly. All figures provided are

well prepared, with readable lettering -- except that the resolution is quite poor.

The English of the manuscript needs, however, some substantial improvement, and, several formulations need to be more precise,

especially already in the abstract. as example, in the title "aluminide" is used (i.e., quite general), and in the abstract, suddenly

AlSiY appears without any further explanation. Please mention precisely in the abstract where you work on!

In the main body of the manuscript, an important issue is the proper formatting of all chemical formulae. Do not forget: This must 

be done everywhere, in Tables, Figures and References. This will improve the readability of the manuscript considerably!

# Line 56: "et al." is Latin, and there is NO dot after "et", but after "al."

# Line 75: When you give a volume, the unit changes accordingly to mm^3

# Line 75/76: How was the coating applied? Sputtering? Electrochemically?

# There should always be a space between a physical quantity and its unit.

# The description of the instrumentation used in this work MUST be improved. Please also mention the respective companies.

# All figure captions should properly describe ALL what one can see in the figure.

# Figure 2 lacks the labelling. Please remove the unreadable text in Fig. 2 (a), and provide better scales in (b)--(d).

# Fig.4: Please explain the dashed and dotted lines in the figure caption!

# Always use the expression "outer layer" intead of "out layer"

Overall, this manuscript provides interesting data material. Thus, it may be published after a thorough revision.

Comments on the Quality of English Language

The English of the manuscript needs some substantial improvement

Author Response

Coatings

Title: In-depth understanding of chemical stripping mechanism of aluminide coatings on nickel superalloys by first-principles calculation

Letter Ref. No.: 2812974

Dear Editor,

Thank you for your January 4, 2024 letter and the reviewers’ comments on our manuscript (coatings-2812974). We found these comments very helpful.

We would like to submit the revised manuscript to Coatings for further review. The changes, deletions and additions have been highlighted in yellow color.

The following summarizes our revisions in response to the reviewers’ criticisms.

We hope you find these revisions acceptable.

Thank you for your kind consideration of our manuscript.

Sincerely,

Chengsong Zhang

Response to reviewer #2:     

 First of all, thank you very much for reviewing our manuscript. Your comments are very significant and professional. In addition, it is very helpful to improve the quality of the manuscript to publish. According to your comments, we answer the following: 

Question 1: The present manuscript is well arranged, and all details of the processes applied are discussed properly. All figures provided are well prepared, with readable lettering -- except that the resolution is quite poor.

Answer 1:

We have checked the resolution of all figures and make sure the resolution of 300 dpi and 1000 dpi for image figures and line figures, respectively.      

Question 2: The English of the manuscript needs, however, some substantial improvement, and, several formulations need to be more precise, especially already in the abstract. as example, in the title "aluminide" is used (i.e., quite general), and in the abstract, suddenly AlSiY appears without any further explanation. Please mention precisely in the abstract where you work on!

Answer 2:

Thank you for your advice. We have check the whole manuscript and replace “aluminide coatings” to “the AlSiY coating” to avoid the confusion on understanding. All the changes have been highlighted by yellow color.

Question 3: In the main body of the manuscript, an important issue is the proper formatting of all chemical formulae. Do not forget: This must be done everywhere, in Tables, Figures and References. This will improve the readability of the manuscript considerably!

Answer 3:

We have checked all chemical formulae and made all format the same. Here we used the plus symbol to connect the chemical formulae to express the mixed chemical removal reagents.  

Question 4: # Line 56: "et al." is Latin, and there is NO dot after "et", but after "al."  

Answer 4:

Thanks for your careful reviewing. We have modified this mistake in the revised manuscript.

Question 5: # Line 75: When you give a volume, the unit changes accordingly to mm^3 

Answer 5:

Thanks for your careful reviewing. We have modified this mistake in the revised manuscript.

Question 6: # Line 75/76: How was the coating applied? Sputtering? Electrochemically?

Answer 6:

The AlSiY coating was deposited on the surface of the K465 superalloy by multi-arc ion plating. The description has been changed and highlighted by yellow color. Thanks for your careful reviewing.

Question 7: There should always be a space between a physical quantity and its unit.

Answer 7

Thanks for your careful reviewing. We have checked and modified those mistakes in the revised manuscript.  

Question 8: The description of the instrumentation used in this work MUST be improved. Please also mention the respective companies.

Answer 8

We have added the details of instruments in the revised manuscript.

Question 9: All figure captions should properly describe ALL what one can see in the figure.

Answer 9

We have checked all figure captions added more explanations in the figure captions to make the captions easier to be understood.

Question 10: Figure 2 lacks the labelling. Please remove the unreadable text in Fig. 2 (a), and provide better scales in (b)--(d).

Answer 10

Thanks for your careful reviewing. We have checked and modified those mistakes in the revised manuscript.

Question 11: # Fig.4: Please explain the dashed and dotted lines in the figure caption!

Answer 11

We have explained the dash line and dot line in the caption of Fig. 4. Thanks for your reminding.

Question 12: # Always use the expression "outer layer" instead of "out layer" 

Answer 12

Thanks for your careful reviewing. We have checked and modified those mistakes in the revised manuscript.  

Reviewer 3 Report

Comments and Suggestions for Authors

This work presents a study of coating on turbines based on aluminum alloys. A coating technology has been developed. The experimental results are summarized and the components of the alloy phases are determined.  Practical conclusions of the application of research coatings have been made.

*The work is written clearly and distinctly.

*Graphs and formulas are clear and informative.

*The conclusions are informative.

*The literature is well selected.

Notes for correction:

*Nickel-based  -   nickel-based ………………………..pk26

*Pay attention to the writing of indices in formulas, example:

Ni3Al - Ni3Al……………………………….pk35

Al2O3  -Al2O3……………………………..pk37

CrO3-   CrO3………..  pk67

HNO3- HNO3………..  pk112

HNO3- HNO3……………..  pk169

ets

Conclusion:

The article In-depth understanding of chemical stripping mechanism of   aluminide coatings on nickel superalloys by first-principles  calculation " can be published after minor corrections.

Author Response

Coatings

Title: In-depth understanding of chemical stripping mechanism of aluminide coatings on nickel superalloys by first-principles calculation

Letter Ref. No.: 2812974

Dear Editor,

Thank you for your January 4, 2024 letter and the reviewers’ comments on our manuscript (coatings-2812974). We found these comments very helpful.

We would like to submit the revised manuscript to Coatings for further review. The changes, deletions and additions have been highlighted in yellow color.

The following summarizes our revisions in response to the reviewers’ criticisms.

We hope you find these revisions acceptable.

Thank you for your kind consideration of our manuscript.

Sincerely,

Chengsong Zhang

Response to reviewer #3:     

First of all, thank you very much for reviewing our manuscript. Your comments are very significant and professional. In addition, it is very helpful to improve the quality of the manuscript to publish. According to your comments, we answer the following: 

Question 1: Notes for correction:

*Nickel-based  -   nickel-based ………………………..pk26

*Pay attention to the writing of indices in formulas, example:

Ni3Al - Ni3Al……………………………….pk35

Al2O3  -Al2O3……………………………..pk37

CrO3-   CrO3………..  pk67

HNO3- HNO3………..  pk112

HNO3- HNO3……………..  pk169

Answer 1:

Thanks for your careful reviewing. We have modified those mistakes in the revised manuscript. 

Reviewer 4 Report

Comments and Suggestions for Authors

This manuscript reports the degradation and refurbishment of aluminide coatings on nickel-based superalloys used in aeroengine turbine components. The study focuses on the chemical stripping process, providing experimental results coupled with first-principles calculations to elucidate the mechanisms involved. Below are some key points that should be considered for review:

1.     Certain sections of the methodology (e.g., first-principles calculations) might benefit from more explicit detailing or references for readers unfamiliar with these techniques. (Clarity on Methodology)

2.     Although the first-principles calculations provide valuable insights, additional validation through more extensive experimental studies or validation against known stripping processes could strengthen the reliability of the findings.

3.     Some figures or tables could be revised for better clarity, ensuring that they effectively support the presented data and conclusions. Also, some words in Figure 3, 5, and 6 are very small.

Author Response

Coatings

Title: In-depth understanding of chemical stripping mechanism of aluminide coatings on nickel superalloys by first-principles calculation

Letter Ref. No.: 2812974

Dear Editor,

Thank you for your January 4, 2024 letter and the reviewers’ comments on our manuscript (coatings-2812974). We found these comments very helpful.

We would like to submit the revised manuscript to Coatings for further review. The changes, deletions and additions have been highlighted in yellow color.

The following summarizes our revisions in response to the reviewers’ criticisms.

We hope you find these revisions acceptable.

Thank you for your kind consideration of our manuscript.

Sincerely,

Chengsong Zhang

Response to reviewer #4:     

First of all, thank you very much for reviewing our manuscript. Your comments are very significant and professional. In addition, it is very helpful to improve the quality of the manuscript to publish. According to your comments, we answer the following: 

Question 1: Certain sections of the methodology (e.g., first-principles calculations) might benefit from more explicit detailing or references for readers unfamiliar with these techniques. (Clarity on Methodology)

Answer 1:

Thanks for your careful reviewing. The theory of first-principles calculations has been carefully described by Ref. [16]. All calculated details and key parameters were all listed in section 2.4. In our present work, the first-principles calculations were conducted on a software platform called Materials Studio. Other researchers no matter who are familiar with these techniques or not could easily conduct the same calculations according to our given parameters. We added more descriptions of calculated slab models in the caption of Fig. 1. 

Question 2: Although the first-principles calculations provide valuable insights, additional validation through more extensive experimental studies or validation against known stripping processes could strengthen the reliability of the findings.

Answer 2:

Yes, I really agree with your opinion. It is the first time we use the first-principles calculation to explain the chemical stripping mechanism of aluminide coatings in the framework of electronic work function theory. There is no related data about both the calculation and the experiment can be found. Therefore, we conducted experiments to verify the correctness of our calculations. In the future, we will carry out more extensive verification works.

Question 3: Some figures or tables could be revised for better clarity, ensuring that they effectively support the presented data and conclusions. Also, some words in Figure 3, 5, and 6 are very small.

Answer 3:

We have modified some figures as your comments. Thanks for your careful reviewing.